# Synthesis of Composites for the Removal of F^-^ Anions

**DOI:** 10.3390/nano13162277

**Published:** 2023-08-08

**Authors:** Adriana Saldaña-Robles, Javier Antonio Arcibar-Orozco, Luz Rocío Guerrero-Mosqueda, César Eduardo Damián-Ascencio, Alfredo Marquez-Herrera, Miguel Corona, Armando Gallegos-Muñoz, Sergio Cano-Andrade

**Affiliations:** 1Department of Agricultural Engineering, University of Guanajuato, Ex Hacienda El Copal km 9, Irapuato 36500, Mexico; rocioguerrerogto@gmail.com (L.R.G.-M.); amarquez@ugto.mx (A.M.-H.); 2Centro de Innovación Aplicada en Tecnología Competitiva, CIATEC, Omega 201, Industrial Delta, Leon 37545, Mexico; jarcibar@ciatec.mx; 3Department of Mechanical Engineering, Universidad de Guanajuato, Salamanca 36885, Mexico; gallegos@ugto.mx (A.G.-M.); sergio.cano@ugto.mx (S.C.-A.); 4Mechanical Engineering and Management, Autonomous University of San Luis Potosi, COARA, San Luis Potosi 78000, Mexico; miguel.corona@uaslp.mx

**Keywords:** fluoride adsorption, graphene oxide composites, nitrogen groups, advanced materials

## Abstract

This work presents the synthesis of amine and ferrihydrite functionalized graphene oxide for the removal of fluoride from water. The synthesis of the graphene oxide and the modified with amine groups is developed by following the modified Hummer’s method. Fourier transform infrared spectrometry, X-ray, Raman spectroscopy, thermogravimetric analysis, surface charge distribution, specific surface area and porosity, adsorption isotherms, and the van’t Hoff equation are used for the characterization of the synthesized materials. Results show that the addition of amines with ferrihydrite generates wrinkles on the surface layers, suggesting a successful incorporation of nitrogen onto the graphene oxide; and as a consequence, the adsorption capacity per unit area of the materials is increased.

## 1. Introduction

The pollution of water bodies around the world represents an issue, mainly due to the presence of hazardous arsenic and fluoride [1,2,3,4,5,6,7]. This is especially hazardous for uses such as water drinking and irrigation in agriculture. Particularly, the presence of fluoride in groundwater is controlled by the mass exchange with the media and mineral composition of the soil [1,8,9].

The presence of fluoride in drinking water within the permissible limits, stablished by the World Health Organization (WHO), i.e., concentrations lower than 1.5 mg L${}^{-1}$, can be beneficial for bone formation [10]. However, in concentrations higher than 1.5 mg L${}^{-1}$, fluoride represents a threat to human health because it reduces the adsorption of calcium and phosphorous, causing dental fluorosis, neural degeneration, and brain damage, among other disorders [11,12,13,14,15]. Thus, there is a need to develop water treatment systems capable of efficiently remove this constituent from groundwater. Some treatments commonly used are precipitation/coagulation [1,13,14,16,17], electrocoagulation [14,18,19]), ion exchange [20,21], membrane processes [22,23], and adsorption processes [22,24,25]. Among these methodologies, adsorption technology is widely used for fluoride removal because its ease of operation, low cost, and high efficiency. Some adsorbents employed are based on innocuous metal oxides such as iron, zinc, aluminum, manganese, and carbon-based materials [12,13,15,18,25,26,27].

Graphene-based adsorbents are novel materials that have attracted much interest in recent years. The diversity of surface oxygenated groups in graphene oxide (GO) makes them ideal for the attraction of adsorbates and the insertion of reactive functional groups [13,20,28,29]. Most of fluoride adsorption occurs through complexation reactions and electrostatic interactions [30]. Thus, if acid groups are present in the GO surface, the adsorption occurs between the metallic complexes in solution and dissociated groups, by electrostatic attraction (such as van der Waals forces) and/or ligand interchanges [31]. In order to improve the adsorptive capacities, different researchers focused on surface modifications for the incorporation of functional groups and other materials, such as metal oxides [3,13,14,15,29,32,33,34,35]. Particularly, iron oxides are widely used to modify the surface of the GO in order to increase the fluoride removal [12,15,29,33]. The efficiency of iron-oxides for removing fluoride depends on physical properties such as crystalline structure, morphology, specific surface area, and particle size. These properties can be modified in the synthesis process [33]. Huang et al. [22] reported that the defluorination mechanism involves the following two categories
(1)$${\mathrm{MO}}_{x}+xH_{2}O\to M{\left(\mathrm{OH}\right)}_{2x}$$
(2)$$M{\left(\mathrm{OH}\right)}_{2x}+yF^{-}\to M{\left(\mathrm{OH}\right)}_{2x-y}F_{y}+y{\mathrm{OH}}^{-}$$
(3)MOH+HF→MOHHF
(4)$$\mathrm{MOHHF}\to \mathrm{MF}+H_{2}O$$
where M represents metals such as Zr, Mg, Fe, Al, etc.

The first category follows Equations (Equation 1) and (Equation 2), where hydrogen bonds hydroxy and fluoride ions. The second category follows Equations (Equation 3) and (Equation 4), where the Fe-doped species are typically hard Lewis acids and possess strong interactions with hard bases of fluoride due to a hydroxy transition. From the above reactions, it is observed that hydroxyl groups play a significant role during fluoride adsorption. Also, the incorporation of nitrogen in the GO surface is an interesting form of surface modification, with the potential of improving during chemical interactions with iron complexes composite formation.

Among nitrogen-based surface modifiers of GO, surfactants improve the adsorption capacity by increasing the surface charge and thus, favoring the electrostatic attraction of inorganic compounds contained in water [20,36,37,38]. Some authors reported that cationic surfactants with ammonium compounds increase the positive zeta potential of the adsorbent, whereas anionic surfactants favor the surface negative charge [39]. Arcibar et al. [40], reported that nitrogen incorporation into the GO matrix has the potential of improving the surface area of composites with iron oxyhydroxides, which is a consequence of a better distribution of iron hexaaquo complexes during nucleation. This effect improves the performance of the composites during the adsorption of surrogates of chemical warfare agents [40]. Also, Mahmudov et al. [41] reported that activated carbon can be functionalized with quaternary ammonium salts, improving the adsorption capacity of perchlorate.

Some composites including graphene oxides functionalized with ferric oxyhydroxide materials have been synthesized, such as composites of magnetic graphene nanoplaquets coated with nanoparticles of FeO-Fe${}_{2}$O${}_{3}$, graphene oxides with reduced Fe${}_{3}$O${}_{4}$ dispersible in water [42], and metal-organic frameworks (MOFs) [43]. The presence of nitrogenated groups has a basic character that promotes electrostatic attraction among the acid groups. This changes the thermodynamic environment, producing new materials characterized by a better dispersion of functional surface groups.

Another potential solution is Capacitive Deionization (CDI), a water desalination technology that has been the focus of recent studies due to its high ion selectivity, efficient water recovery, and low energy consumption. However, its industrial application has yet to be realized due to several limitations, including restricted electrosorption capacity, slow electrosorption rate, and poor cycling stability [44]. The photo-electric capacitive deionization (PECDI) technologies are presented as a novel variant of the CDI technologies, which takes advantage of photo-enhancement of ionic transport kinetics, which is enabled by the edge-enhanced vertical graphenes, obtaining higher adsorption capacities as well as faster responses. However, this technology is still under investigation and its full potential and limitations are still to be understood [45].

To the best knowledge of the authors, only some materials based on graphene oxides as a matrix for ferric oxyhydroxides that has not been functionalized with amine groups have been studied [13,33]. The present work focuses on GO modified with nitrogen and ferrihydrite composites for the removal of fluoride from aqueous solutions. We study the impregnation of GO with nitrogen compounds, i.e., amines with different pK${}_{a}$ prior to iron nucleation. The goal is to determine if the addition of nitrogen moieties with different chemistry can improve the nucleation of iron during ferrihydrite formation and in this way, obtain advanced adsorbent materials.

## 2. Materials and Methods

### 2.1. Reagents

Graphite powder (99%) and sodium fluoride (NaF) (purity: 99.95%) were obtained from Sigma Aldrich. All chemical reactants were of analytical grade, such as sulfuric acid (H${}_{2}$SO${}_{4}$), nitric acid (HNO${}_{3}$), potassium permanganate (KMnO${}_{4}$), potassium hydroxide (KOH), sodium hydroxide (NaOH), hydrogen peroxide (H${}_{2}$O${}_{2}$, 30%), carbonyldiamine (CH${}_{4}$N${}_{2}$O), diphenylamine (C${}_{12}$H${}_{11}$N), ethylene glycol (C${}_{2}$H${}_{6}$O${}_{2}$), ferric nitrate nonahydrate (Fe(NO${}_{3}$)${}_{3}\cdot$9H${}_{2}$O), deionized water and TISAB II.

### 2.2. Experimental Set-Up

The synthesis of the GO composites functionalized with ferrihydrite/amine groups was developed in three steps:

Step 1: GO synthesis. The synthesis of the GO was carried out according to the modified Hummer’s method [46], where 10 g of graphite powder was mixed with 230 mL of H${}_{2}$SO${}_{4}$ (98%). The mixture was stirred for 2 h into a cold bath. After this, 30 g of KMnO${}_{4}$ were added to the mixture and it was stirred for 180 min and kept at a temperature below 20 °C until it reached 2 °C. Then, the mixture was aged at room temperature for 30 min. After that, 250 mL of deionized water was added slowly under a cold bath and stirred for 15 min. Subsequently, 1.4 L of deionized water was added, followed by 100 mL of H${}_{2}$O${}_{2}$ at 30%. The mixture was aged 12 h at room temperature. This mixture was washed several times with deionized water to obtain a pH of 3.03 approximately. Then, the mixture was centrifuged at 5000 rpm for 30 min, and the graphite oxide obtained was cooled to −20 °C for 15 days. Finally, the graphite oxide obtained was sonicated for 2 h to exfoliate it and obtain graphene oxide.

Step 2: Addition of amine groups. The addition of amine groups was developed following two methodologies. The former is carried out by modification with carbonyl diamine (urea) and the latter with diphenylamine. For both methods, 198 mL of deionized water was added to 1.2 g of GO, and the mixture was sonicated by 2 h in order to disperse GO into water. Then, either 3.6 g of carbonyl diamine or 3.6 g of diphenylamine were dissolved in 210 mL of ethylene glycol, added to the GO dispersion, and stirred at 140 rpm and 80 °C for 24 h. Finally, both modified mixtures were centrifuged at 400 rpm for 40 min, washed with water-ethylene glycol (1:1), and centrifuged several times until a pH of 7 was reached in the residual solution. These mixtures were filtered (Watman, 40) and dried at 80 °C for 15 h. The resultant materials were named GOU and GODA because of the addition of amine groups through the carbonyl diamine and diphenylamine, respectively.

Step 3: GOU and GODA decorated with ferrihydrite (FH) (Fe${}_{5}$HO${}_{8}\cdot$ 4H${}_{2}$O). First, 1 g of GOU (GODA) was dispersed into 165 mL of deionized water by sonication for 2 h. Then, 39.13 g of Fe(NO${}_{3}$)${}_{3}$ 9H${}_{2}$O was added to the GOU (GODA) dispersion, 800 mL of deionized water was added, the mixture was stirred at 100 rpm, and then 0.1 M of KOH was dropped carefully until a pH of 7.6–8 is reached. After that, the mixture was centrifuged at 4500 rpm for 10 min. The product was dispersed in water and centrifuged several times in order to remove all unreacted residues. Finally, the products were dried in a convection oven at 60 °C for 24 h, and the resulting materials were named GOUFH and GODAFH.

## 3. Characterization

### 3.1. Fourier Transform Infrared Spectrometry

The Fourier transform infrared spectrometry (FT-IR) was obtained in a Thermo Scientific NICOLET IS10 in attenuated total reflectance (ATR) mode in order to characterize the functional groups. The spectrum was generated in the range of wave numbers between 4000 and 550 cm${}^{-1}$ with 32 scans and 4 cm${}^{-1}$ of resolution.

### 3.2. X-ray

The X-ray diffraction (XRD) was carried out in a Philips X’pert spectrometer with a 0.02° step at 30 mA and 40 kV between anglles of 10° and 90°. The X-ray analysis is developed with a κ−α X-ray like excitation source of 1486.6 eV. The analysis was developed in the range of 292–288 eV, and the spectra data is deconvoluted.

### 3.3. Raman Spectra Microscopy

The Raman spectra microscopy was carried out in a RENISHAW (In-Via), using a laser wavelength of 514 nm as excitation source, and a laser spot of 1 μm with a power at the sample below 10 mW. The exfoliation of graphitic oxide into GO monolayers was carried out through a SONIC VCX 750 model (20 kHz, 750 W) in a direct immersion oven of titanium.

### 3.4. Thermogravimetric Analysis

The thermogravimetric analysis (TGA) of the sample was carried out with a thermogravimetric differential scanning calorimetry analysis (TGA; NETZSCH, STA449F3), from room temperature to 800 °C, with a temperature increasing rate of 10 °C min ${}^{-1}$ and 25 mL min${}^{-1}$ of air as heat carrier.

### 3.5. Surface Charge Distribution

The distribution of surface charge was obtained by adding 100 mg of the compound to 50 mL of NaCl 0.1 M electrolyte solution. The suspension was bubbled with N${}_{2}$ by 2 h. Then, 0.1 M HCl was added to the suspension. After that, the suspension was titrated with NaOH in a 916 Ti-touch (Metrohom). Finally, the distribution of surface charge was obtained by determining the characterization of the surface of activated carbons in terms of their acidity constant contributions.

### 3.6. Specific Surface Area and Porosity

The specific surface area and porosity of the compounds were measured in a Micromeritics TriStar II Plus 2.03 equipment by N${}_{2}$ adsorption at −196 °C, before the sample was degassed at 60 °C and a vacuum of 10${}^{-4}$ atm, for 2 h. The calculation of the specific surface area and pore volume was developed using the Brunauer-Emmet-Teller (BET) and Barret-Joiner-Halenda (BJH) relations, respectively.

### 3.7. Scanning Electron Microscopy

Materials were observed by Scanning Electron Microscopy (SEM) in a Helios microscope (FEI 600 NANOLAB) at an acceleration voltage of 8 keV, using secondary and backscattered electrons. Before the analysis, the samples were grinded in an agate mortar, then they were suspended in isopropanol. Finally, the samples were mounted in a SEM-pin.

### 3.8. Adsorption Isotherms

*Step 1.* The adsorption experimental study for the removal of fluoride ions was carried out at 25 °C and a pH of 4. First, 0.025 g of each synthesized material (GO, GOFH, GODA, GOU, GODAFH, GOUFH) as adsorbent was placed into an Erlenmeyer flask. Then, 20 mL of a fluoride solution at different initial concentration was added (0–10 m L${}^{-1}$). The flask with the mixture was stirred at 160 rpm. The pH was adjusted with 0.1 M NaOH and HCl. Finally, the mixture was centrifugated at 4000 rpm for 15 min and the solution was collected and analyzed by a fluoride ion selective electrode (Orion Star A214 Thermo Scientific). The adsorption capacity was adjusted using both, the Langmuir and Freundlich models. From these results, the better adsorbent was selected for further characterization.

*Step 2.* For the selected material, the adsorption experimental studies for the removal of fluoride ions were carried out at 25, 35, and 45 °C, and a pH of 4, 6, and 8. First, 0.025 g of the adsorbent was placed into an Erlenmeyer flask. Then 20 mL of a fluoride solution at different initial concentrations (0–10 mg L${}^{-1}$) was added. These mixtures were shaken at 160 rpm, at different temperatures, and at different pHs. The pH was adjusted during the experiment with 0.1 M NaOH and HCl until the equilibrium was reached. Then, the mixtures were separated by centrifugation at 4000 rpm, for 15 min. Finally, the clear solutions were collected, and the fluoride residual concentration was analyzed. The adsorption capacity was adjusted to the Langmuir and Freundlich models.

The Langmuir model implies a monolayer capacity and is given as
(5)$$q_{e}=\frac{q_{L}bC_{e}}{1+bC_{e}}$$
where $q_{e}$ is the equilibrium adsorption capacity, $C_{e}$ is the equilibrium solute concentration, $q_{L}$ is related to the maximum adsorption capacity, and *b* is related to the bonding energy of adsorption.

The Freundlich model, which represents heterogeneous adsorption, is given as
(6)$$q_{e}=K_{F}C_{e}^{1/n}$$
where $K_{F}$ is the adsorption equilibrium constant representative of the adsorption capacity, $C_{e}$ is the equilibrium adsorption capacity, and *n* is the adsorption intensity.

### 3.9. The Van’t Hoff Equation

Considering the relations between the Gibbs free energy and the equilibrium constant ($\Delta G^{0}=\Delta H^{0}-T\Delta S^{0}$ and $\Delta G^{0}=-RT\ln K_{d}$), the van’t Hoff equation is given as
(7)$$\ln K_{d}=-\frac{\Delta H^{0}}{R}\left(\frac{1}{T}\right)+\frac{\Delta S^{0}}{R}$$
where *T* is the temperature in absolute units, *R* is the ideal gas constant and K${}_{d}$ is the equilibrium constant. The adsorption constant was determined from the data coming from the isotherm by the identification K${}_{d}=b$, and in consequence K${}_{d}$ is dimensionless and its numerical value is equal to *b* [47]. The absolute value of $\Delta G^{0}$ is associated to specific interactions. For instance, $\Delta G^{0}$ around 4 kJ/mol is associated to a hydrophobic force whereas a value ranging from 4 to 10 kJ/mol is associated to Van der Waals forces.

## 4. Results and Discussion

### 4.1. Fourier Transform Infrared Spectrometry

Base material: The infrared curve for the synthesized GO contains a region between 3000–3600 cm${}^{-1}$ where a wide band is present, which corresponds to both adsorbed water and also bulk hydroxil groups. This water accumulation increases the inter-planar distances, producing a structure modification, and through exfoliation in the synthesis process for the GO material, it generates imperfections in the hexagonal lattice, such as vacancies. These imperfections generate an anchorage of the epoxy functional groups between the basal planes and carboxylic acid in the edge of the sheets [48,49].

*Modified materials:* A modification of the GO surface suggests the reduction of these functional groups, favoring the formation of bonds with the nitrogenated groups as well as the oxygen surface groups of iron. In the same way, it is possible to identify in GO the presence of carbonyl (C=O) at ∼1730 cm${}^{-1}$, carboxyle (COOH) at ∼2360 cm${}^{-1}$, epoxide at ∼1201 cm${}^{-1}$, and C−O groups at ∼1034 cm${}^{-1}$. The above-mentioned groups obtained in the GO fix its hydrophilic properties. These bands are in agreement with those of the materials reported in the technical literature [46]. As is observed the bands at 1150 cm${}^{-1}$ and 1600 cm${}^{-1}$ reveal the presence of urea whereas the amine group can be attributed to the band at 3400 cm${}^{-1}$.

The composites named GOFH, GODAFH, and GOUFH, prevailed in three regions which are shown in Figure 1. In the spectrum, it is observed the bands at ∼3150 cm${}^{-1}$ which represent the stretching of O−H bonds characterized by a wide band, are related to structural hydroxide and adsorbed H${}_{2}$O. Also, the bands between 1650 and 1300 cm${}^{-1}$ belong to symmetric and asymmetric stretching of the C−O and the deformation of water at ∼1630 cm${}^{-1}$. In addition, the stretching of Fe−O is shown in the bands at 980, 890, 617, and 565 cm${}^{-1}$. However, the lower crystallinity of the ferrihydrite is a factor that affects the infrared spectrum, decreasing the resolution of the signals [50,51].

Kuang et al. [33] synthesized GO impregnated with goethite and akageneite for the remotion of fluoride. They found that the synthesized materials have characteristic signals at ∼3300 cm${}^{-1}$ due to the stretching of structural hydroxile groups, at ∼1628 cm${}^{-1}$ due to C=O groups, at ∼1620 cm${}^{-1}$ due to vibrations of C=C bounds, at ∼1380 cm${}^{-1}$ due to carboxile groups, and at ∼1032 cm${}^{-1}$ due to the stretching of C−O. Also, the composite material GO/Goethite, shows a signal at ∼3340 cm${}^{-1}$ related to the stretching of hydroxile group of the goethite, confirming the presence of Fe−O with signals at 847 and 671 cm${}^{-1}$. For the composite material GO/Akageneite the FT-IR shows signals at 588 and 466 cm${}^{-1}$. These signals are in agreement with the signals shown in Figure 1, related with oxygenated groups for the GO and some signals related to stretching of Fe−O bounds in the range of 400–600 cm${}^{-1}$.

### 4.2. X-ray Analysis

Figure 2 shows the X-ray diffractogram of the oxidized material synthesized by the modified Hummers method. In the GO, it is observed a characterized signal at an angle of 7° (2θ), which is related to the distance between layers of GO, and indicates the positive oxidation of graphene. An absence of a pronounced peak at 26.5° is also observed, suggesting that all the graphite has been converted into GO. This peak is observed for the modified materials only. The results obtained in the DRX analysis are in agreement with those reported in previous works and exhibit the transformation of graphite to graphene oxide [52,53].

It is also observed that the ferrihydrite diffractogram shows noise related to its lower degree of crystallinity. The most defined bands at 35 and 63 can be attributed to 2-line ferrihydrite. These results are in agreement with the reported in the literature [55,56]. In general, it is observed an amorphous compound with wide peaks related to both, a lower degree of crystallinity and a small size of crystals (see Figure 2). The identity of the ternary composite was confirmed in similar patterns for both GOFH and GOUFH materials, with the two characteristic bands centered at 35° and 63°. The small difference in the peaks observed at 23, 30, and 40° could be related to the presence of urea [57]. However, these differences are too small to be analyzed.

### 4.3. Raman Analysis

Raman spectroscopy is a non-destructive test useful to analyze the structure of materials with carbon content. The Raman analysis was carried out for the characterization of the graphene oxides synthesized in the present work, i.e., GO, GOUFH, and GODAFH, and is shown in Figure 3. The Raman test shows a characteristic peak at 1336, 1292, and 1331 cm${}^{-1}$ for the GO, GOUFH, and GODAFH materials, respectively. This band is known as the D band, which is an indicator of the presence of imperfections in the graphite crystalline structure, such as order, bound angle, vacancies, and lattice defects. Also, the bands observed at 1574 and 1567 cm${}^{-1}$ in GO and GODAFH materials, respectively, correspond to a dispersion of first order in the E2g modes. Both sets of bands have a similar intensity due to the strong oxidation produced by the modified Hummers method, which induces a structural disorder. Additionally, the small peaks at 2627 and 2673 cm${}^{-1}$ for the GOUFH and GOUFG materials are related to the resonance of the D band, which is absent in the GODAFH material. The double resonant modes in the Raman dispersion at the 2322 cm${}^{-1}$ band for the GOUFH are still a matter of discussion; however, some authors [58,59,60] mention that these bands are related to a process of two phonons of the D and D’ type. This process involves the contribution of optic phonons (D) and acoustic longitudinal phonons (D’), which are usually known as G” (or D+D’) modes.

### 4.4. Thermogravimetric Analysis

Figure 4 shows the TGA termograms of the GOUFH material. Figure 4a shows three weight loss stages. It is observed that the lower weight loss of 7.04% occurs from 32 °C to 100 °C, is related to either desorption or evaporation of water from dehydroxylation of hydrous mixed oxide and GO of the composite. This is in agreement with the results reported in the literature [15,16]. The weight loss of 13.08% from 100 to 400 °C is related to the decomposition of nitrogen groups in the GOUFH material. In addition, it is observed a weight loss of 2.81% from 400 to 500 °C. Thus, a total weight loss of 23.83% was observed. The high temperature required for the decomposition of the GOUFH, indicates that after the functionalization with amine groups, its thermal stability is increased.

To determine the nature of the DTGA profile, the rate of change of the weight loss with respect to time is considered, as is shown in Figure 4b. It is observed that the DT spectrum has two endothermic peaks at just before 48 °C and 395 °C, and two exothermic bands at around 222 °C and 310 °C.

As is observed in Figure 4 the GO materials have lower thermal stability since it is completely degraded at temperatures of 190 °C whereas the modified material degrades at higher temperatures.

### 4.5. XPS Analysis

The functional groups on the adsorbent surface of GOUFH and GO were analyzed by the XPS technique. Figure 5a shows the XPS spectra for the GOUFH, which is the material with the best F${}^{-}$ adsorption capacity. In the deconvoluted spectra, C1s is observed because the spectrum of the sample contains three peaks at 284.2, 285.4, and 288.2 eV, corresponding to the carbonyl, epoxide/ether, and carboxyl functional groups, respectively. These results confirmed the presence of GO with a considerable oxidation degree [61]. A section of the XPS not deconvoluted is shown in the insert of Figure 5a. It is observed a peak of low intensity at 400 eV, is characteristic of N1s [62]. This N1s peak is related to the amine groups added during the modification of the GO surface. These results indicate that the GOUFH maintains its functionality even after the composite formation.

Figure 5b shows that the O1s region could be deconvoluted into three overlapped peaks at 531, 530, and 529 eV, corresponding to Fe(OH)${}_{3}$, Fe(OH)O, and iron oxides in the lattice structure, respectively. These results are in agreement with those reported in the literature [16,33]. In addition, the GOUFH shows the formation and growth of iron oxide on its surface, as expected.

Figure 5c displays the C1s XPS spectra for the graphene oxide (GO) sample, which consists of three peaks at 283.3, 284.4, and 287.0 eV. These peaks correspond to the C−C, C−O, and C=O functional groups, respectively. Despite GO has the potential to exhibit six possible C1s components like graphene, the XPS spectrum of graphene oxide typically presents two main peaks, in contrast to the single peak displayed by graphene, as depicted in Figure 5a,c. These observations confirm the formation of GO [61,63,64].

Figure 5d exhibits the O1s XPS spectra for the graphene oxide (GO) sample, containing three peaks at 531.8, 532.4, and 532.9 eV. These peaks correlate with the C−OH, C−O, and C−C functional groups, respectively (3). It’s significant to note that the O1s XPS spectra for the GOUFH material (Figure 5b) demonstrate lower binding energy than the O1s XPS spectra for the OG sample, as indicated in Figure 5c (dashed line, non-normalized curve).

### 4.6. Surface Charge Distribution

The charge distribution of the synthesized materials is shown in Figure 6 through the pH${}_{\mathrm{PZC}}$ for the GODA, GOU, GODAFH, and GOUFH materials. As it is observed, the GODA and GOU materials have a pH${}_{\mathrm{PZC}}$ of 5.8 and 7.2, respectively, whereas the GODAFH and GOUFH materials have a pH${}_{\mathrm{PZC}}$ of 8.5 and 8.3, respectively. The difference between the first materials (GODA and GOU) with respect to the second materials (GODAFH and GOUFH) is due to the addition of ferric oxyhydroxides groups, which affect the surface charge density. Also, the surface charge of the GODAFH and GOUFH materials depends of the pH, because at a pH < pH${}_{\mathrm{PZC}}$, the adsorbent materials have a positive surface charge, whereas at a pH > pH${}_{\mathrm{PZC}}$, the adsorbent materials have a negative surface charge. Thus at a pH < pH${}_{\mathrm{PZC}}$, the surface of the GODAFH and GOUFH materials is more suitable for the electrostatic attraction of F${}^{-}$. This mechanism has already been described in the technical literature [33,65]. As is known, the charge distribution on the surface of the materials is related to the dissociation and deprotonation of surface groups. Thus, if the pH${}_{\mathrm{PZC}}$ of the GODAFH and GOUFH materials is higher than the pH${}_{\mathrm{PZC}}$ of ferrihydrite, the nucleation, and condensation of iron complexes on the GO surface are linked to a higher amount of hydroxyl groups of an acidic nature.

The pH${}_{\mathrm{PZC}}$ obtained for each synthesized material was lower than 3 for the GO and around 9 for the FH material and around 8.6, 7.2, 5.8, 8.5, and 8.3 for the GOFH, GODA, GOU, GODAFH, and GOUFH respectively. As is known, the modification of GO with nitrogenated groups affects the surface charge of the adsorbent material [20,39]. Thus, the difference in the pH${}_{\mathrm{PZC}}$ between the GOUFH and the GODAFH is related to the fact that the GOUFH is synthesized with a primary amine, where its basic character promotes a bigger electrostatic attraction with the cation acid groups, i.e., the hydroxyl groups of the ferric oxyhydroxides. On the other hand, the GODAFH material was synthesized with a secondary amine, which limits the electrostatic attraction with the acidic hydroxyl groups of the ferric oxyhydroxide. This is reflected in the specific surface area i.e., the GOUFH shows a bigger specific surface area than the GODAFH.

### 4.7. BET Analysis

The specific surface area is determined from the nitrogen adsorption isotherms, shown in Figure 7. The GO shows a specific surface area of 4 m${}^{2}$ g${}^{-1}$ which is less than the surface area reported in the literature of 20–40 m${}^{2}$ g${}^{-1}$ [46,66]. This discrepancy is due to the stiffness and beds of the laminar structure, which affects the laminar spacing of the GO, not allowing the nitrogen molecule to go into the laminar structure. On the other hand, after the modification of the GO surface with amine groups, the specific surface area obtained is 4 and 5 m${}^{2}$ g${}^{-1}$ for the GOU and GODA materials, respectively. It is observed that these values did not represent a significant change with respect to that of GO.

The composites modified with ferric oxy-hydroxides show a significative increase in the specific surface area, i.e., 247, 226, and 239 m${}^{2}$ g${}^{-1}$ for the GOFH, GODAFH, and GOUFH materials, respectively. This increase in the specific surface area indicates a successful growth of iron oxides on the surface of the GO since iron oxyhydroxides are known to have a large surface when in an amorphous structure. This growth depends on different synthesis factors, such as the flow rate at which the ferric solution is added to the GO and the stirring rate. Also, these specific surface areas are in agreement with the values reported in the literature [67]. Table 1 shows the specific surface area and pore size distribution of the materials. It is observed that the GOUFH has a bigger specific surface area than the GODAFH. In the same manner, the pore diameter of the GOUFH is bigger than that of the GODAFH, suggesting a higher adsorption capacity for the GOUFH. This result is also in agreement with the reported in the technical literature [68].

The pore size distribution of the synthesized materials is shown in Figure 8. This distribution is obtained using the Barrett–Joyner–Halenda (BJH) method, assuming that the pore filling due to condensation represents a well-defined interface in the pores. Both materials, GOUFH and GODAFH, show the maximum pore volume with pore diameters around 3.155 and 3.303 nm respectively. That is, narrow mesopores are present, which is in agreement with the reported in the literature, i.e., iron oxides have an average pore diameter between 3 and 4 nm [69]. Also, the GODAFH shows the highest pore volume, which has a shoulder at a pore width of 4 nm.

### 4.8. SEM Analysis

SEM images of the materials are shown in Figure 9. It is observed that the GO morphology displays the distinctive sheet-like of graphene, which is corrugated by the effect of the oxygen groups. Interestingly, the GOU seems to have a more corrugated and folded surface. According to the GO model described by Szabó et al. [70], the corrugated carbon network has a ribbon-like arrangement linked by a periodically cleaved ribbon of cyclohexane chairs. The slight tilting angle between the boundaries of the regions explains the wrinkling of the layers.

The successful incorporation of nitrogen, i.e., amines, on the GO allows an electrostatic attraction of OH-NH groups, causing a slight change in the torsion angle, and increasing the wrinkles on its surface. The GODAFH and GOUFH materials show the characteristic particles of ferrihydrite without a particular morphology. A fluffy cluster of particles seemed to be the result of the agglomeration of nanoparticles for the GODAFH material.

### 4.9. Adsorption Isotherms

Adsorption isotherms were developed to determine the fluoride adsorption capacity of the synthesized materials. Figure 10 shows the characterization of the interactions between F${}^{-}$ and the adsorbent materials, i.e., GO, GOU, GODA, GOFH, GOUFH, and GODAFH. The parameters used for the Langmuir and Freundlich models are given in Table 2. It is observed that the regression coefficient of the Freundlich model is bigger than that of the Langmuir model. This suggests an uneven distribution of the adsorption energy sites and a possible formation of adsorbate multilayers.

It is observed, that the adsorption capacity of the GOUFH and GODAFH is considerably higher than that of the GO materials, which is due to the presence of ferrihydrite. This can be linked to the slight differences induced for the two methods for nitrogen incorporation on GO. The lack of fluoride capacity on the GO materials can be attributed to the negatively charged sites of acidic moieties that serve as repulsion sites of F${}^{-}$ ions. When considering a fluoride equilibrium concentration of 0.5 mg L${}^{-1}$, the Freundlich constant capacity of the GO, GODA, and GOU, is 0.919, 0.539, and 0.574 (mg g${}^{-1}$) (L mg${}^{-1}$)${}^{1/n}$, respectively. At this same concentration, the Freundlich constant capacity of GODAFH and GOUFH is 9.345, and 10.033 (mg g${}^{-1}$) (L mg${}^{-1}$)${}^{1/n}$, respectively. When considering the adsorption capacity per surface area at an equilibrium concentration of 0.5 mg L${}^{-1}$, the adsorption capacities were 0.0276 and 0.0281 mg m${}^{-2}$ for the GODAFH and GOUFH materials, respectively. These values represent a better surface area coverage than that of the GOFH material, i.e., 0.0293 mg m${}^{-2}$. This suggests that an increase in the adsorption capacity is related to a better exposition of coordination sites when nitrogen is present. As discussed above, amine groups cause torsion of the graphene layers, allowing for the exposition of reactive groups during iron nucleation. Thus, the resultant material has a better exposition of iron exaaquo complexes and, as a consequence, a slightly higher adsorption capacity. This idea is strengthened when considering that the “n” parameter of the Freundlich model has a higher value for the composites containing nitrogen. For instance, the area changes as the surface is deformed due to the presence of functional groups. Considering the functional
(8)$$A\propto \int_{R}d^{2}\sigma \sqrt{\det\gamma }$$
where $\det\gamma =\det P\left[g\right]$ is the determinant of the pullback of the metric which is obtained as $\gamma_{\alpha \beta }=\frac{\partial x^{a}}{\partial \sigma^{\alpha }}\frac{\partial x^{b}}{\partial \sigma^{\beta }}g_{ab}$, where sum over repeated indexes is understood, and the integration region *R* is determined by the boundaries of a selected region of the GO layers with characteristics lengths $l_{x}$ and $l_{y}$ which are greater to zero. In the absence of functional groups, the graphene layers are nearly flat and the metric describing the layers is approximately $\delta_{ab}$. Thus, the surface area is proportional to
(9)$$A_{0}\propto \int_{R}d^{2}\sigma$$

This case implies that a flat 2D surface embedded in the Euclidean 3D space is considered. The pullback is trivial since detγ=1. Now, any deformation of the flat surface can be expressed as a deformation of the embedding coordinates, namely, the $x^{i}\in \left\{x,y\right\}$. Now with the deformation
(10)$$x^{i}=x^{i}+f\left(x^{i}\right)\;,$$
thus, the A functional changes as follows
(11)$$A^{\prime }\propto A_{0}+l_{y}\left|f\left(x\right)\right|+l_{x}\left|f\left(y\right)\right|+\left|f\left(x\right)f\left(y\right)\right|\;,$$
which implies that any deformation of the flat region *R* contains positive definite terms that increase the area of the surface. Thus, it is concluded that the wrinkles generated by the addition of functional groups always tend to increase the surface area and thus increasing the posible cites of adsorption.

The adsorption tests for the synthesized materials are obtained at a pH 4 because almost all the materials show a pH${}_{\mathrm{PZC}}$ higher than 5.8. Thus, at a higher pH (see Figure 6), the surface of the materials is positively charged and the adsorption capacity is drastically reduced. This may be the reason that the GOUFH material displayed a bigger adsorption capacity than the GODAFH material. It is probably related with the presence of NH${}_{2}$ groups contained in the urea, which triggers the formation of anchorage (for iron clusters) and adsorption sites, as opposed to the difenilamine which contains NH groups.

It is also observed that for the Freundlich model, the 1/n parameter obtained for the GO, GODA, and GOU materials is 0.979, 0.959, and 0.915, respectively. This suggests a lower homogeneity of adsorbed F${}^{-}$ for the GO materials. However, the composite materials functionalized with ferrihydrite displayed a higher adsorption capacity of F${}^{-}$, i.e., with 1/n values of 0.747, 0.576, and 0.575 for the GOFH, GODAFH, and GOUFH respectively.

The GO material modified with urea and ferrihydrite, named GOUFH, shows a higher adsorption capacity than the other modified materials. Thus, it is used as a reference to test the adsorption properties at several conditions. The adsorption isotherms where obtained for the GOUFH material at a temperature of 25 °C, 35 °C, and 45 °C, and at a pH of 4, 6, and 8 (see Figure 11).

The experimental data obtained is predicted using both, the Freundlich and Langmuir models. The correlation coefficient, $R^{2}$, is used to compare the models. It is observed that both, the Freundlich and Langmuir models, predict well the experimental data. The parameters used in the models to predict the adsorption of F${}^{-}$ in the GOUFH material at the conditions established are given in Table 3.

The adsorption experiments were obtained at a pH smaller than the pH${}_{\mathrm{PZC}}$, which is about 8.3 (see Figure 6). It is observed that the GOUFH is capable to remove well the fluorides for a pH lower than 8. However at a pH of 8, the pH${}_{\mathrm{PZC}}$ is quite close and the material is not able to adsorb F${}^{-}$; as a consequence, the R${}^{2}$ is decreased. This is caused by the addition of ferrihydrite as well as the nitrogenated groups contained in the urea deposited on the GO surface. Since the GOUFH is positively charged, it increases the electrostatic force between the fluoride and the surface of the material. It is also observed that the modified GO increases its adsorption capacity because of the presence of Fe, which has a high affinity with the F${}^{-}$ and, thus, increases the probability of adsorption.

The adsorption isotherms for a pH 4 at a temperature of 35 °C and 45 °C show a better prediction using the Langmuir model with a $q_{L}$ of 10.147 and 11.018 mg g${}^{-1}$, respectively. On the other hand, the isotherm at a pH of 4 and a temperature of 25 °C shows the highest adsorption, and the Freundlich model shows a better prediction than the Langmuir model. The $K_{F}$ of 8.697 mg g${}^{-1}$ and 1/n of 0.502 are related to a higher adsorption intensity. Thus, the modifications of the GO surface with oxihydroxides and amine groups from the urea suggest an increase in the active sites on the surface of the ternary material. As is expected, the pH of 4, which is below the pH${}_{\mathrm{PZC}}$, i.e., ∼8, allows increasing the adsorption capacity. This favors the protonated sites, increasing the F${}^{-}$ retained on the surface of the material.

In order to evaluate the accuracy of the predictions, standardized residuals (SR) were employed. Standardized residuals are defined as the ratio of the residuals over the estimated variance. In the present work, the SR ranges from −2.0 to 2.0 and is distributed randomly around the abscissa. This result suggests that there is no over or under-estimation of the overall prediction. Overall, the SR suggests that the model fits reasonably well with the measured values.

There exist materials based on nano-functionalized graphene oxide (GO) for fluoride removal. Some of these materials include GO/Al${}_{2}$O${}_{3}$[71] with an adsorption capacity of 4.68 mg g${}^{-1}$, and GO doped with zirconium with an adsorption capacity of 6.12 mg g${}^{-1}$[72]. Another material, Zr-MCGO, reported an adsorption capacity of 8.84 mg g${}^{-1}$[73]. These materials displayed lower adsorption capacities than the reported in the present work, which was around 11 mg g${}^{-1}$. This increase in the adsorption capacity is due to the folds produced during the functionalization of the GO, which is reflected in an increase in surface area.

However, multi-functionalized materials with greater removal capacities have also been reported. These include the Zr-Chitosan membrane and GO with an adsorption capacity of 29.05 mg g${}^{-1}$[14], FeOOH+Ac/GO with an adsorption capacity of 19.82 mg g${}^{-1}$[33], α−FeOOH/rGO with an adsorption capacity of 24.67 mg g${}^{-1}$, and HIAGO with an adsorption capacity of 27.8 mg g${}^{-1}$[15]. Also, MgO/MgFe${}_{2}$O${}_{4}$/GO [9] has an adsorption capacity of 34 mg g${}^{-1}$. It should be noted that HIAGO, MgO/MgFe${}_{2}$O${}_{4}$/GO, as well as FeOOH+Ac/GO, has dual metal functionalization, which implies a greater number of functional groups on the surface, thereby increasing the adsorption capacity.

### 4.10. Effects of pH and Temperature

The results show that higher adsorption capacities are obtained at a pH of 4. This is related to the protonation of the surface below the pH${}_{\mathrm{PZC}}$ as well as the natural effect of the electrostatic interactions. This suggests that, as the pH increases, a decrease in the adsorption capacity is obtained [74]. Thus, it is interesting to compare adsorption capacities close to the pH${}_{\mathrm{PZC}}$ of the materials. At this point, the electrostatic interactions are decreased to zero, and the fluoride adsorption is driven by a different mechanism. The particular interaction can be due to ligand exchange reactions of the form
(12)$$\equiv \mathrm{Fe}-\mathrm{OH}\;+F^{-}\to \equiv \mathrm{Fe}-F+{\mathrm{OH}}^{-}\;.$$

This process releases OH groups to the solution, leading to a change in the surface charge distribution, promoting electrostatic interactions and repulsions. Also, a considerable disruption of the adsorption isotherms is observed at a pH of 8. It is known that iron oxides are amphoteric; thus, they can be dissolved in acidic and basic media. It is possible that at this pH some of the iron nuclei, particularly those bonded with the graphene oxide surface, are dissolved, leading to a dissolution of the reactive material. Hence, the adsorption capacity is decreased. It is also important to mention that early studies on ferrihydrite have reported the formation of colloidal ferrihydrite, which is complicated to separate from dissolved Fe [75].

In summary, the maximum adsorption capacities were obtained at a temperature of 25 °C. It is also observed that the adsorption capacity decreases as the temperature increases. This suggests that an exothermic process is favored at low temperatures. This phenomenon is usually attributed to an increase in the internal energy of adsorbed molecules that overcome the adsorption energy.

### 4.11. The Van’t Hoff Equation

In Figure 12 it is presented the regression analysis of the experimental data with the non-linear van’t Hoff equation. As is observed the K${}_{d}$ constant has a non-linear dependence with 1/T at values of pH of 6 and 8, whereas it shows almost a linear dependence with 1/T for a pH of 4. Besides, it is observed a negative entropy change of entropy at pH of 4 and 6, i.e., $\Delta S^{0}=-62.36$ kJ kmol${}^{-1}$ K${}^{-1}$ for a pH of 4 and $\Delta S^{0}=-140.68$ kJ kmol${}^{-1}$ K${}^{-1}$ for a pH of 6. However, the entropy change was positive for values of pH of 8, i.e., $\Delta S^{0}=129.40$ kJ kmol${}^{-1}$ K${}^{-1}$ for a pH of 8. Thus, it is inferred that the adsorption sites are decreasing for pH of 4 and 6 whereas they are increasing at a pH of 8. In addition, the changes in enthalpy have a positive change for all the values of pH studied in the present work, suggesting exothermic reactions.

## 5. Conclusions

In the present work, amine-graphene oxide-ferrhydrite materials were sinthesized, and used as adsorbents of fluoride ions from an aqueous solution. The synthesis was developed following the modified Hummers method. The characterization of the material was carried out by means of Fourier transform infrared spectrometry, X-ray, Raman spectroscopy, thermogravimetric analysis, surface charge distribution, specific surface area and porosity, adsorption isotherms, and the van’t Hoff equation.

The difractogram shows that the materials contain characteristic peaks, which confirms the material structure. Also, the addition of amines is corroborated by the increase of wrinkles on the surface layers of the modified materials, suggesting a successful incorporation of nitrogen into the graphene oxide. This causes an increase in the adsorption capacity per unit area of the materials. In addition, the SEM analysis confirms an amorphous surface of the materials, which is a characteristic of ferrihydrite particles. Thus, the incorporation of nitrogen into the GO matrix shows an improvement of the surface area due to the better distribution of iron hexaaquo complexes during the nucleation.

Besides, the presence of the nitrogenated groups shows a basic character, promoting the electrostatic attraction among the acid groups contained in cations, changing the thermodynamic environment, and giving room to new materials characterized by a better dispersion of functional groups on the surface.

The adsorption tests confirm that the materials modified with nitrogen increase their adsorptive capacities as well as the surface area. This can be related to a better exposition of coordination sites when nitrogen is present. Particularly, the GOUFH material, which is synthesized with a primary amine, shows the best F${}^{-}$ adsorption capacity, maintains its functionality even after the composite formation, shows the formation and growth of iron oxide on its surface, and shows a bigger specific surface area.

Finally, the Freundlich and Langmuir models, as well as the linear van’t Hoff equation predict well the F${}^{-}$ adsorption capacity of the materials.

## Figures and Tables

**Figure 1 nanomaterials-13-02277-f001:**
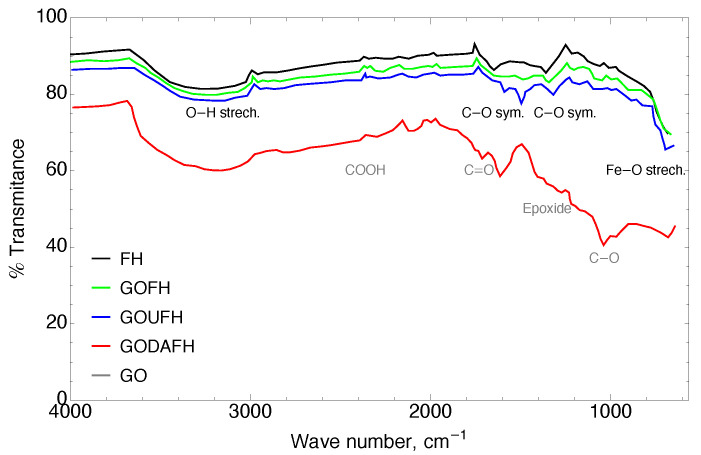
FT-IR of the FH (black), GOFH (green), GOUFH (blue), GODAFH (red), and GO (gray) materials.

**Figure 2 nanomaterials-13-02277-f002:**
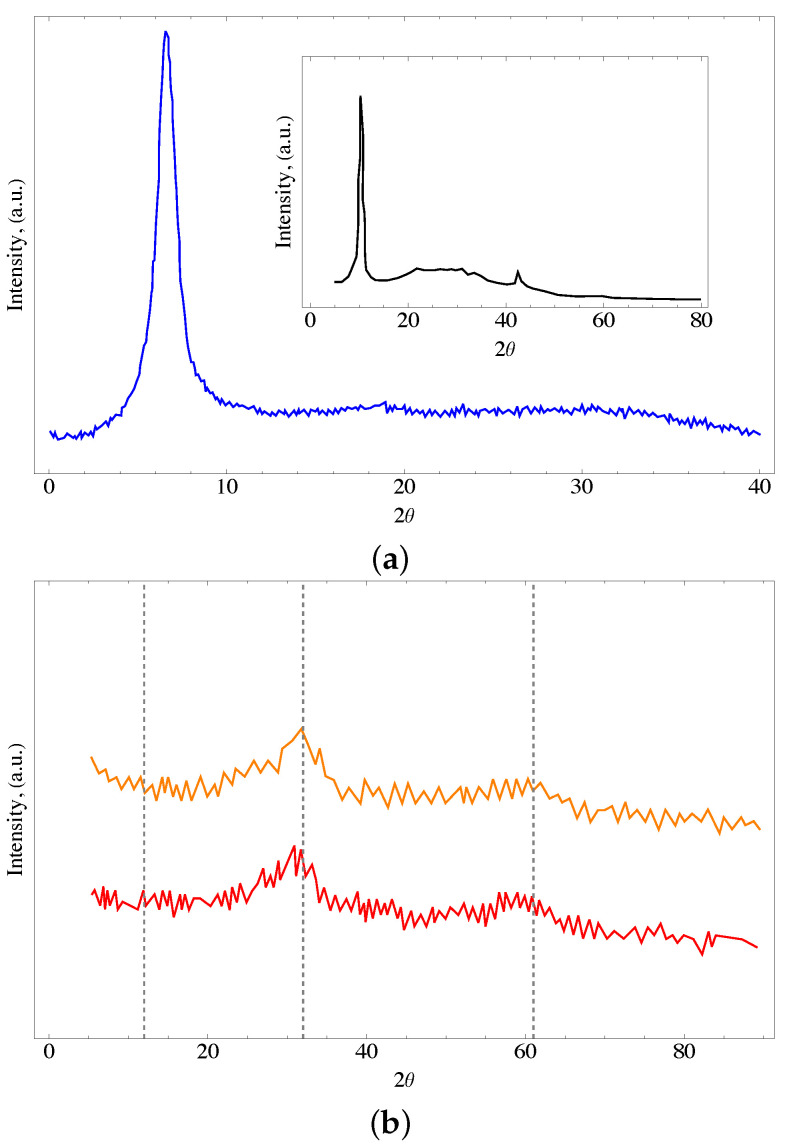
X-ray of the (**a**) GO where the insert shows the graphene oxide reported in [54] and (**b**) for the GOFH (orange line) and GOUFH (red line) materials.

**Figure 3 nanomaterials-13-02277-f003:**
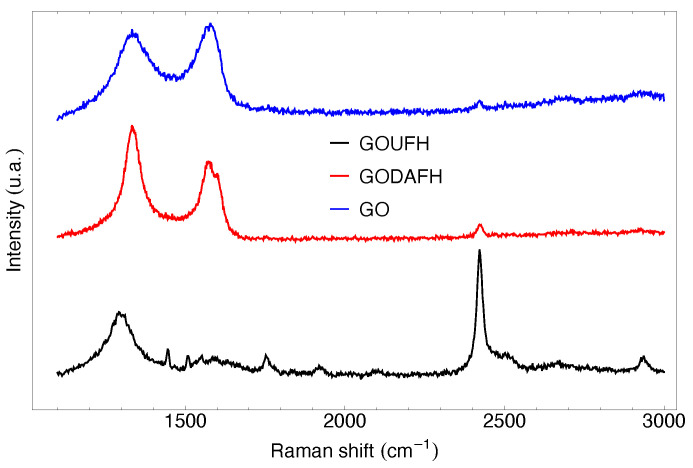
Raman spectroscopy of the GO (blue), GODAFH (red), and GOUFH (black).

**Figure 4 nanomaterials-13-02277-f004:**
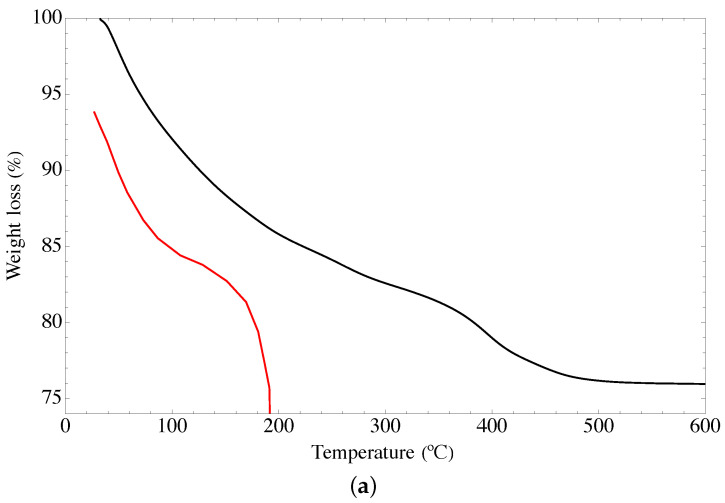
TGA spectra of GOUFH material (**a**) weight loss and (**b**) rate of change of the weight loss, the red line corresponds to the precursor materials and the black line to the GOUGFH material.

**Figure 5 nanomaterials-13-02277-f005:**
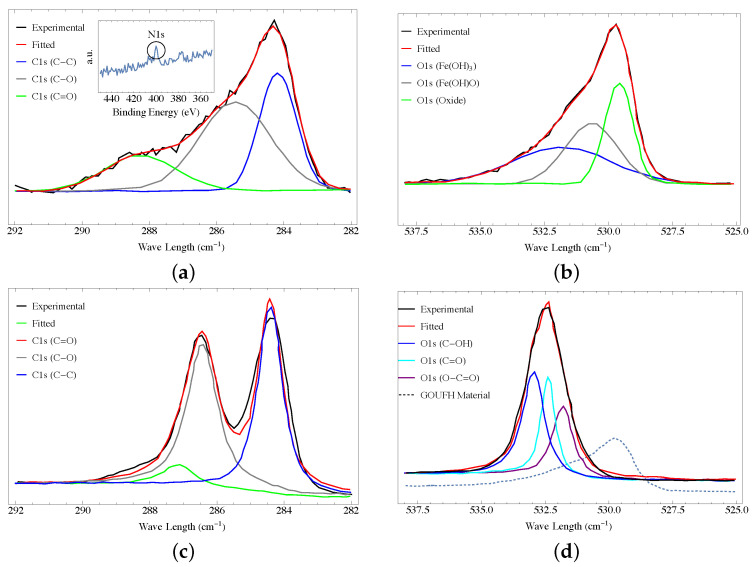
XPS spectra of the GOUFH material (**a**) C1s and (**b**) O1s, and for the precursor (**c**) C1s and (**d**) O1s.

**Figure 6 nanomaterials-13-02277-f006:**
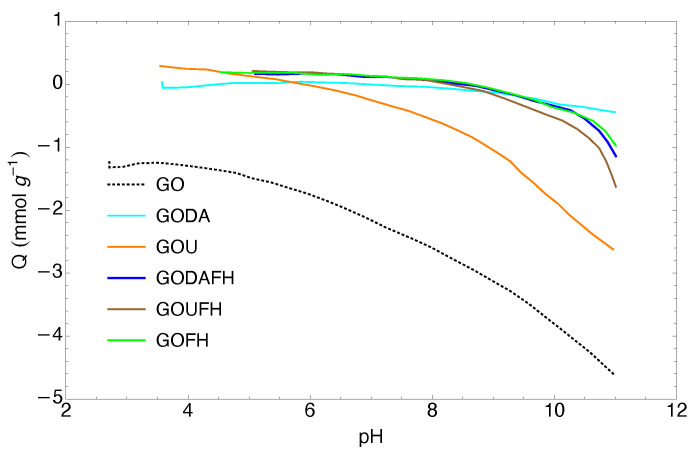
Charge distribution of the synthesized materials.

**Figure 7 nanomaterials-13-02277-f007:**
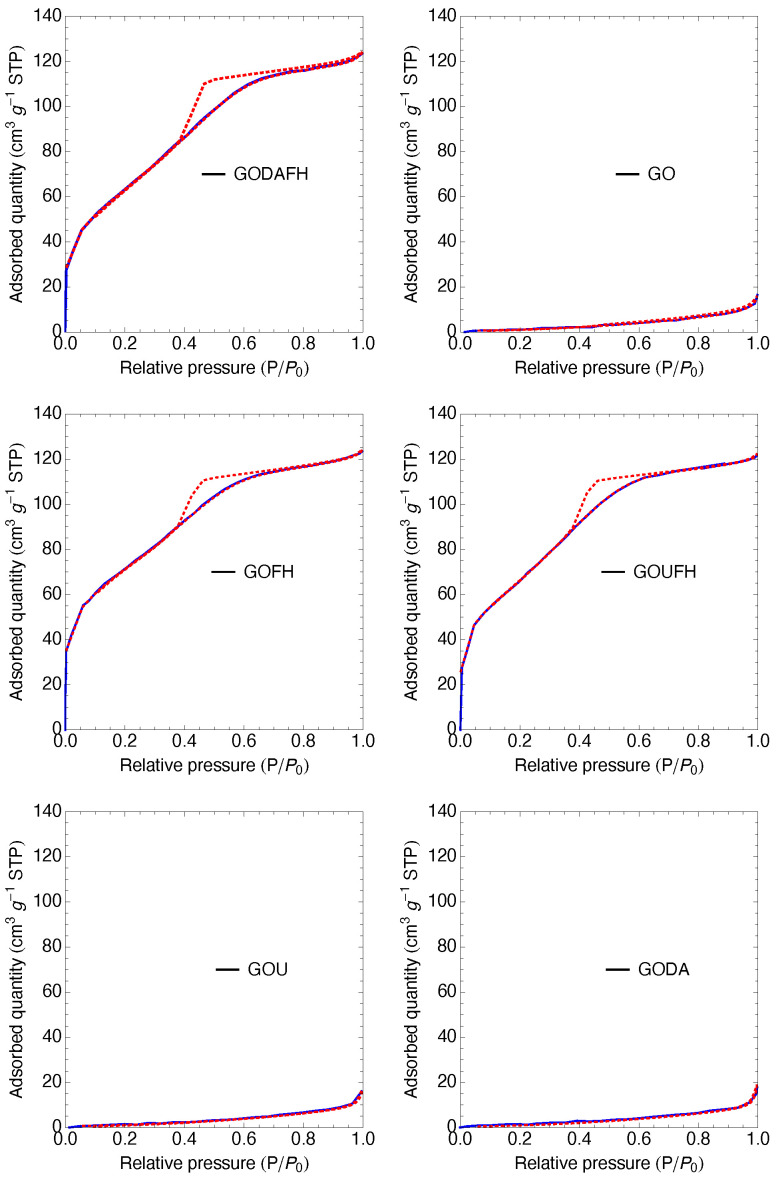
Nitrogen adsorption isotherms of the synthesized materials, The blue line represents the adsorption process, while the red dashed line indicates the adsorption-desorption process.

**Figure 8 nanomaterials-13-02277-f008:**
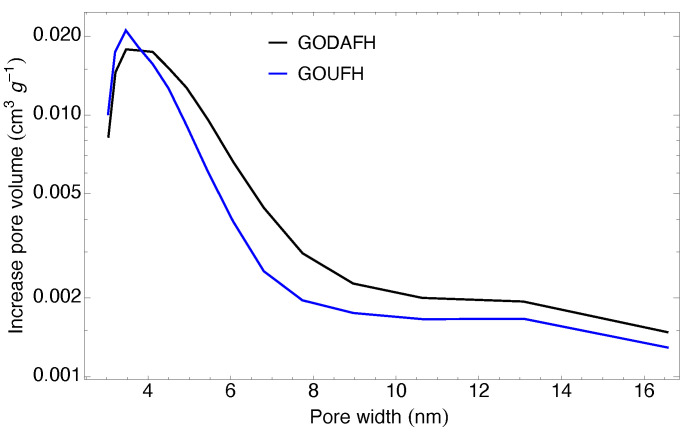
Pore size distribution obtained with the BJH method for the GOUFH (blue line) and the GODAFH (black line) materials.

**Figure 9 nanomaterials-13-02277-f009:**
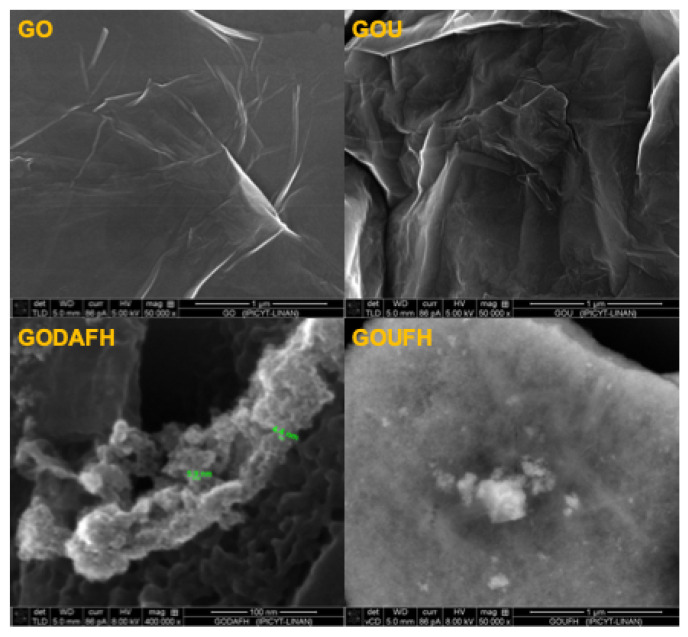
SEM images of the materials.

**Figure 10 nanomaterials-13-02277-f010:**
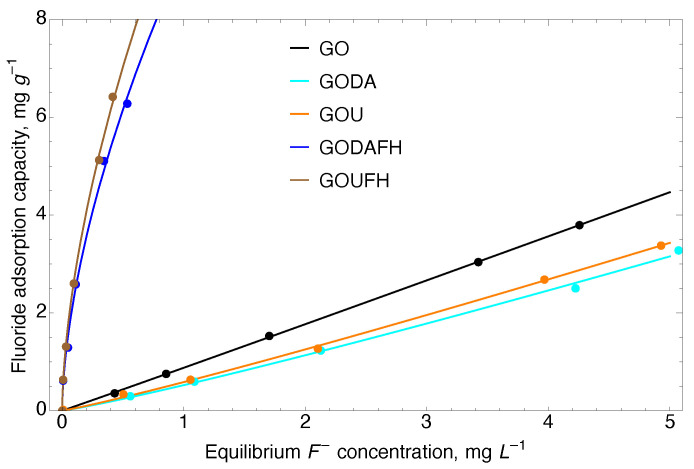
Adsorption isotherms of fluoride adsorption on the studied material at 25 °C and a pH 4.

**Figure 11 nanomaterials-13-02277-f011:**
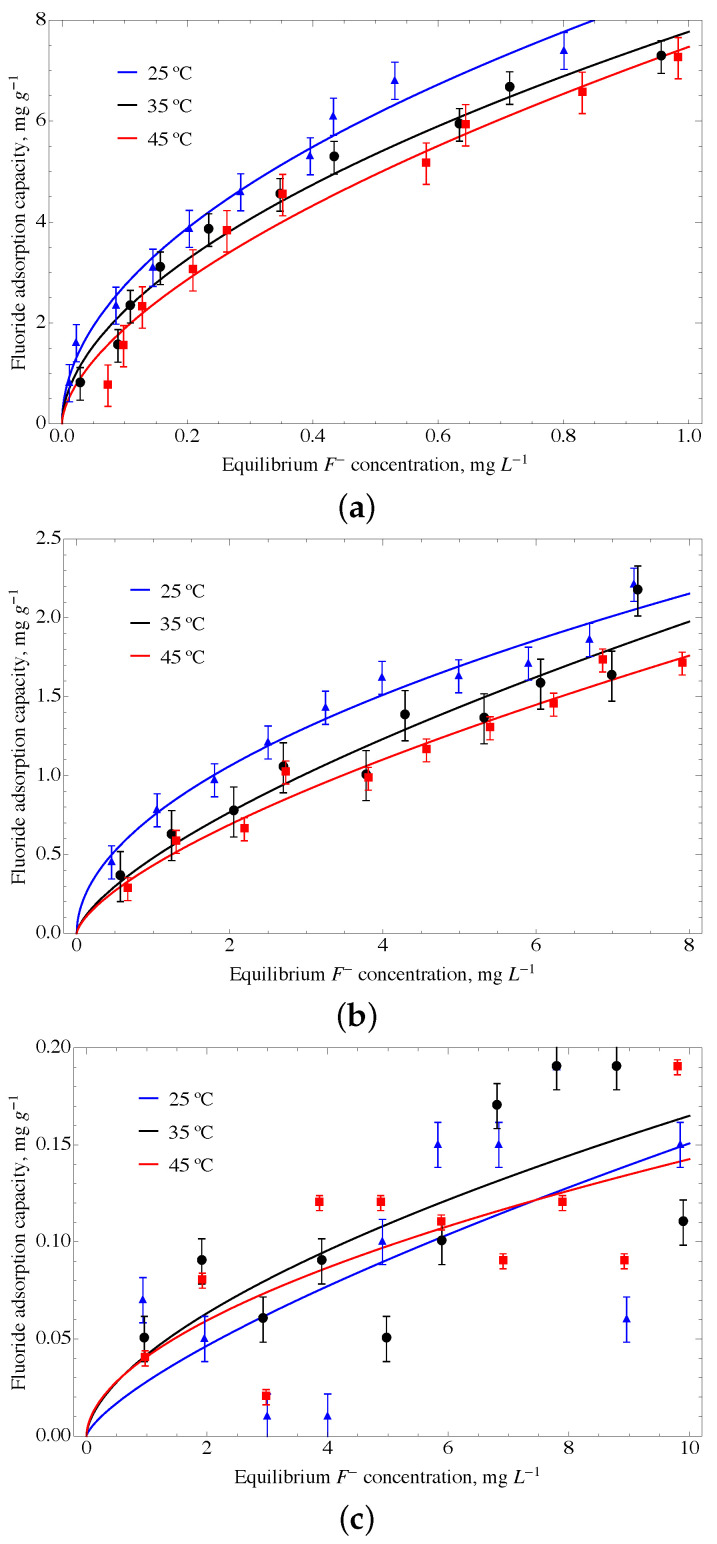
Adsorption isotherms obtained using the Freundlich model for the GOUFH material at a temperature of 25 °C, 35 °C and 45 °C, and at a pH of (**a**) 4, (**b**) 6, and (**c**) 8.

**Figure 12 nanomaterials-13-02277-f012:**
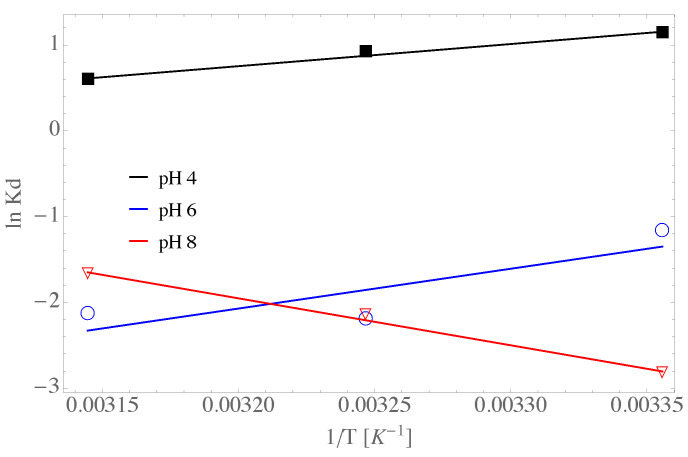
Linear van’t Hoff prediction for the GOUFH material.

**Table 1 nanomaterials-13-02277-t001:** Specific surface area and pore size distribution.

Material	Specific Surface Area	Volume	Pore Diameter
	(m${}^{2}$ g${}^{-1}$)	(cm${}^{3}$ g${}^{-1}$)	(nm)
GO	4	0.015	5.594
GODA	4	0.012	5.826
GOU	5	0.012	5.668
GOFH	247	0.185	3.207
GODAFH	226	0.185	3.303
GOUFH	239	0.183	3.155

**Table 2 nanomaterials-13-02277-t002:** Parameters of the Freundlich and Langmuir isotherms at a pH of 4 and a temperature of 25 °C for the synthesized materials.

		Freundlich		
**Material**	**K${}_{F}$** **(mg g** ${}^{-1}$ **) (L mg** ${}^{-1}$ **)** ${}^{1/n}$	**1/n**	**R${}^{2}$**	**SE**
GO	0.919	0.979	0.99	0.031
GODA	0.539	0.959	0.99	0.078
GOU	0.574	0.915	0.99	0.053
GODAFH	9.345	0.576	0.99	0.304
GOUFH	10.033	0.575	0.99	0.327
		**Langmuir**		
**Material**	**q${}_{L}$** **mg g** ${}^{-1}$	**b** **L mg** ${}^{-1}$	**R${}^{2}$**	**SE**
GO	2.007	32.705	0.98	0.035
GODA	1.002	35.290	0.98	0.119
GOU	1.014	45.411	0.98	0.118
GODAFH	9.405	3.524	0.98	0.223
GOUFH	9.706	4.176	0.97	0.493

**Table 3 nanomaterials-13-02277-t003:** Parameters used in the Freundlich and Langmuir models to predict the adsorption of F${}^{-}$ on the GOUFH material at a temperature of 25 °C, 35 °C and 45 °C, and a pH of 4, 6, and 8.

		Freundlich		
		**pH = 4**		
**Temperature** **°C**	$K_{F}$ **(mg g** ${}^{-1}$ **) (L mg** ${}^{-1}$ **)** ${}^{1/n}$	1/n	$R^{2}$	**SE**
25	8.687	0.502	0.984	0.301
35	7.772	0.540	0.981	0.323
45	7.472	0.597	0.970	0.420
		pH = 6		
25	0.745	0.510	0.970	0.099
35	0.479	0.682	0.922	0.160
45	0.432	0.676	0.964	0.097
		pH = 8		
25	0.028	0.730	0.362	0.054
35	0.042	0.595	0.510	0.041
45	0.041	0.543	0.490	0.490
		**Langmuir**		
		**pH = 4**		
**Temperature** **°C**	$q_{L}$ **(mg g** ${}^{-1}$ **)**	B **(L mg** ${}^{-1}$ **)**	$R^{2}$	**SE**
25	10.325	3.117	0.979	0.430
35	10.147	2.497	0.992	0.206
45	11.018	1.806	0.983	0.309
		pH = 6		
25	2.854	0.308	0.960	0.116
35	4.120	0.110	0.909	0.180
45	3.520	0.117	0.958	0.107
		pH = 8		
25	0.405	0.059	0.367	0.054
35	0.303	0.116	0.503	0.041
45	0.213	0.187	0.473	0.037

## Data Availability

Not aplicable.

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
