# Peer review of "Synthesis of Composites for the Removal of F^-^ Anions"

_nanomaterials, 2023, doi:10.3390/nano13162277_

Round 1
Reviewer 1 Report
It is an interesting study, with relevant results in a subject of great interest. The work is complete and well strutuctured which is worthy to be published after minor revision:
- There are many writing mistakes, the following are some examples:
In line 108 of page 3 it is said: “The first method is carried out by modification with carbonyl diamine”. Needless to say how is carried out the second method.
Line 156 of page 4: “The specific surface area and porosity of the compounds were measured in a Mi- cromeritics TriStar II Plus 2.03 equipment by N2 adsorption at −196 ◦C before the sample was degassed at 60 ◦C and a vacuum of 10−4 atm, for 2 h”. The measurements are performed after degassing, therefore a comma is missing “−196 ◦C, before”.
The unity is missing in figure 2.
Which is the meaning of “is a characteristic where the graphite is not present”?
Caption of figure 5 needs to indicate what is a) and b).
Line 296 of page 9: “results, indicate” Comma is left over.
Line 300 of page 9: “results are agreement”, should be “results are in agreement”
Line 352 of page 10: the the is duplicated.
Figure 9 has two dots. Spare a point
The same content is repeated:
“It is observed, that the adsorption capacity of the GOUFH and GODAFH is consider able higher than that of the GO materials, which is due to the presence of ferrihydrite. It s also observed that the GOUFH and GODAFH materials displayed a higher adsorption capacity than the GO materials.”
Line 288 of page 8, it is said: “The functional groups on the adsorbent surface were analyzed by the XPS technique”. As only GOUFH was analysed it should say: “The functional groups on the adsorbent surface of GOUFH were analyzed by the XPS technique”
- ¿What does it correspond to the intense peak at 26.5 º?
- It needs to be better explained why these huge differences of surface areas, from 4-5 to more than 200 m2/g.
- Adsorption results must be compared with those of bibliography.
As stated above minor corrections are needed.
Author Response
The pdf with a detailed response to the reviewers is added.

Reviewer 2 Report
In the present work, amine-graphene oxide-ferrhydrite materials were sinthesized, and used as adsorbents of fluoride ions from aqueous solution. The adsorption tests confirm that the materials modified with nitrogen increase their adsorptive capacities as well as the surface area. This can be related to a better exposition of coordination sites when nitrogen is present.
(1) The TGA termograms of GO and GODAFH nanomaterials are also suggested for comparison with GOUFH to demonstrate the improved thermal stability.
(2) XPS of GO is needed for comparison with GOUFH.
(3) Why authors not provide the pore size distribution with only the nitrogen adsorption isotherms.
(4) In addition to XPS test, zero potential tests also help to understand the surface charges introduced by functional group decorations.
(5) Authors are suggested to discussion the possibility of selective adsorption of GOUFH when immersed in multi-kind of anions (e.g., F- and Cl-).
(6) How about the adsorption performance compared to previous studies.
(7) The electro-sorption technology can also absorb the cation or anions in drinking water or wastewater and seawater (Energy Storage Materials 50 (2022) 395–406). Especially, photo-electric capacitive deionization has been proposed by Yang H. C. (e.g., Chemical Engineering Journal 422 (2021) 130156). Authors are suggested to comment on this in the introduction.
(8) some mistakes like “aqueou solution”
minor
Author Response

(The authors gave the same response as above.)

Round 2
Reviewer 2 Report
Authors have addressed the comments well